

# Ten years of temperature and wind observation on a 45-m tower at Dome C, East Antarctic plateau

Christophe Genthon[1], Dana Veron[2], Etienne Vignon[1], Delphine Six[3], Jean-Louis Dufresne[1], Jean-Baptiste Madeleine[1], Emmanuelle Sultan[4], François Forget[1]

[1]Laboratoire de Météorologie Dynamique, Paris, France
[2]School of Marine Science and Policy, Newark, USA
[3]Institut des Géosciences de l'Environnement, Grenoble, France
[4]Museum National d'Histoire Naturelle, Paris, France

*Correspondence to*: Christophe Genthon (christophe.genthon@cnrs.fr)

**Abstract.** Long-term, continuous in situ observations of the near-surface atmospheric boundary layer are critical for many weather and climate applications. Although there is a proliferation of surface stations globally, especially in and around populous areas, there are notably fewer tall meteorological towers with multiple instrumented levels. This is particularly true
in remote and extreme environments such as the Eastern Antarctic plateau. In the article, we present and analyze 10 years of data from 6 levels of meteorological instrumentation mounted on a 42-m tower located at Dome C, East Antarctica near the Concordia research station, producing a unique climatology of the near-surface atmospheric environment (Genthon et al., 2021,a,b). Monthly temperature and wind data demonstrate the large seasonal differences in the near-surface boundary layer dynamics, depending on the presence or absence of solar surface forcing. Strong vertical temperature gradients (inversions)
frequently develop in calm, winter conditions, while vertical convective mixing occurs in the summer leading to near-uniform temperatures along the tower. Seasonal variation in wind speed is much less notable at this location than the temperature variation as the winds are less influenced by the solar cycle; there are no katabatic winds as Dome C is quite flat. Harmonic analysis confirms that most of the energy in the power spectrum is at diurnal, annual and semi-annual time scales. Analysis of observational uncertainty and comparison to reanalysis data from ERA-5 indicate that wind speed is
particularly difficult to measure at this location. Data are distributed on PANGAEA data repository, see data availability section.

## 1 Introduction

Antarctica is a land of extremes. In terms of climate, the Antarctic continent is where some of the coldest in situ surface temperatures and largest surface wind speeds have been measured. The high Antarctic plateau has long been renowned for its
frequent and extreme surface-based temperature inversions (Phillpot and Zillman (1970), Zang et al. (2011)), inspiring studies that 1) deepen understanding of polar boundary layer physics (van de Wiel et al. (2017), Baas et al. (2018), Abraham & Monahan (2019), Kayser et al. (2020)) and 2) assess model simulation (Bazile et al. (2014), Couvreux et al. (2020),



Vignon et al. (2017), van der Linden et al. (2019)) of the very stable atmospheric boundary layer. However, because both the environment itself and the logistics needed to access and work in such an environment are challenging, long continuous

time-series of meteorological observations in this region are sparse and mostly confined to near-surface information. Networks of automatic weather stations (AWS), including those managed by the Antarctic Meteorological Research Center (Bromwich and Stearns (1993), Colwell et al. (2016)), report air temperature and wind at one level within a few meters of the surface. Some of the AMRC stations were deployed in the early 1980s, providing data that extend over 5 decades. The longest continuous meteorological observations occur near occupied scientific stations, the most extensive ones from stations

that were established during the International Geophysical Year 7 decades ago. Such multidecadal data have allowed Antarctic-wide estimations of surface climate trends from in situ reports (e.g. Steig et al. (2009)). Radiosondes launched at many of the manned stations have led to evaluations of variability and trends in the surface and free atmosphere over the last decades (e.g. Marshall (2002), Ricaud et al. (2020b)) but they are "snapshot" observations at a fixed time of the day. They do not provide information about the large diurnal variations that characterize the surface atmosphere. In addition, there are

errors in these observations due to the relatively long response time in cold environments (Hudson et al. (2004), Tomasi et al. (2011)). Radiosondes transit the near-surface atmosphere, where much of the diurnal and vertical variations occur, in only a few seconds, too fast for the sondes to fully adjust to the environment (Genthon et al. 2010).

There is need for long time-series of atmospheric boundary layer properties to assess and improve model performance near

the surface. Although surface-based remote sensing techniques exist to sample certain aspects of the Atmospheric Boundary layer (ABL) (e.g. Argentini et al. (2005, 2014), Petenko et al. (2019), Ricaud et al. (2020a)), first order variables such as temperature and wind are better characterized with in situ sensors, which can provide more accurate and better resolved data. However, in situ ABL measurements require infrastructure such as masts or towers. There are few places on the Antarctic plateau equipped with a tower, most located in close proximity to stations occupied year-round. For example, Hudson and

Brandt (2005) reported the presence of a surface-based temperature inversion using observations from a 22-m tower at the South Pole (Amundsen-Scott station). The tallest tower on the Antarctic plateau suitable for meteorological profiling stands at Dome C on the eastern Antarctic plateau (Genthon et al. 2013). The permanently occupied Concordia station employs staff that provide maintenance and service even in winter. This is crucial because the instruments operate in extreme conditions that potentially affect optimal measurement such as frost deposition where layers of frost must be manually removed in order

to ensure correct operation of the instrument. Dome C is at high elevation (more than 3200 m above sea-level), and situated more than 1000 km inland from the coast. The surface is permanently snow covered. Thus, both the surface albedo and the surface emissivity are high, and the atmosphere above is cold and dry, providing favorable conditions for the occurrence of strong, near-surface temperature inversions, particularly in winter when the sun is below the horizon (polar night). In contrast, in summer there is a long period during which the sun is always above the horizon with a diurnally varying

elevation angle. Shallow convection can occur when the sun is highest (during the 'day') (Argentini et al. (2005), Genthon et al. (2010)), alternating with periods when an inversion builds and then dissipates when the sun is lower on the horizon





(during the 'night'). Therefore, Dome C is a perfect location to observe the stable atmospheric boundary layer, from extreme cases in winter to daily variations in summer and transition with convection, and thus provide data to evaluate theory and models in a large range of polar ABL stability cases.


In this paper, we present 10 years of in situ temperature and wind observations at Dome C from 2010 to 2019 at 6 observation levels distributed along a ~42-m tower that are part of the CALVA-ACDC (in situ data for CALibration – VAlidation of meteorological and climate models and satellite retrievals, from Antarctic Coast to Dome C, acronym generally shortened to CALCA) project. It has been more than 10 years since the tower is erected and equipped with

meteorological sensors, and the first paper describing the meteorological system has been published (Genthon et al. 2011). Some measurements have been adapted and improved and the dataset has grown considerably longer, making analyses of interannual features of the Dome C near-surface atmosphere possible. The aim of the present paper is therefore twofold: i) to describe a 10-year temperature and wind dataset acquired along the Dome C meteorological tower; and ii) to perform a first climatological analysis of the intra- and inter-annual variability of the temperature and wind at 6 levels in the Dome C ABL.

The paper also aims to invite anyone interested to proceed with further analysis and exploitation of the data which are made available on public repository (Genthon et al. (2021,a,b). The observation system is described in section 2, along with a discussion of data quality and limits. The 10-year record is presented and analyzed for ABL features such as variability, extremes and trends in section 3, and includes statistics and extremes of the temperature inversion. In section 4, the observations are used to evaluate the latest generation of ECMWF (European Center for Medium-range Weather Forecasts)

reanalyses, ERA5 (Hersbach et al., 2020). This is important as reanalyses tend to be used as observation surrogates when and where observations are most acutely missing, such as in Antarctica. However, this is also where meteorological analyses may be most questionable due to limited observational constraints and to more limited evaluation of physical processes in the extreme Antarctic environment. Discussion and conclusions close this paper in section 5.

## 2. Setting, instruments, data and methods

Fig. 1 shows a topographic map of Antarctica with the location of Dome C (Dome Charlie) indicated. Dome C is located at 123° 21' E, 75° 06' S, 3233 m above sea level. Concordia (location C, fig. 1) is a Franco-Italian research station, permanently occupied since 2005. A tower erected in 2004 and located roughly 700 m southeast of Concordia station, stands upwind of the main wind flow direction. Initially, the tower rose 30-m above the surface; the height was extended to 45 m in 2008. An initial suite of meteorological instruments was deployed at 6 levels along the 45-m tower just after the tower expansion

(Genthon et al. 2011, 2013). However, due to snow accumulation of ~8 cm per year, the top observation height is currently less than 45 m above the snow surface (Genthon et al. 2015). For example, the full tower reached only to 44.7 m above the surface in 2008. As there has been no height extension since, the height of the tower top above the snow surface has gradually decreased by ~8 cm per year on average.


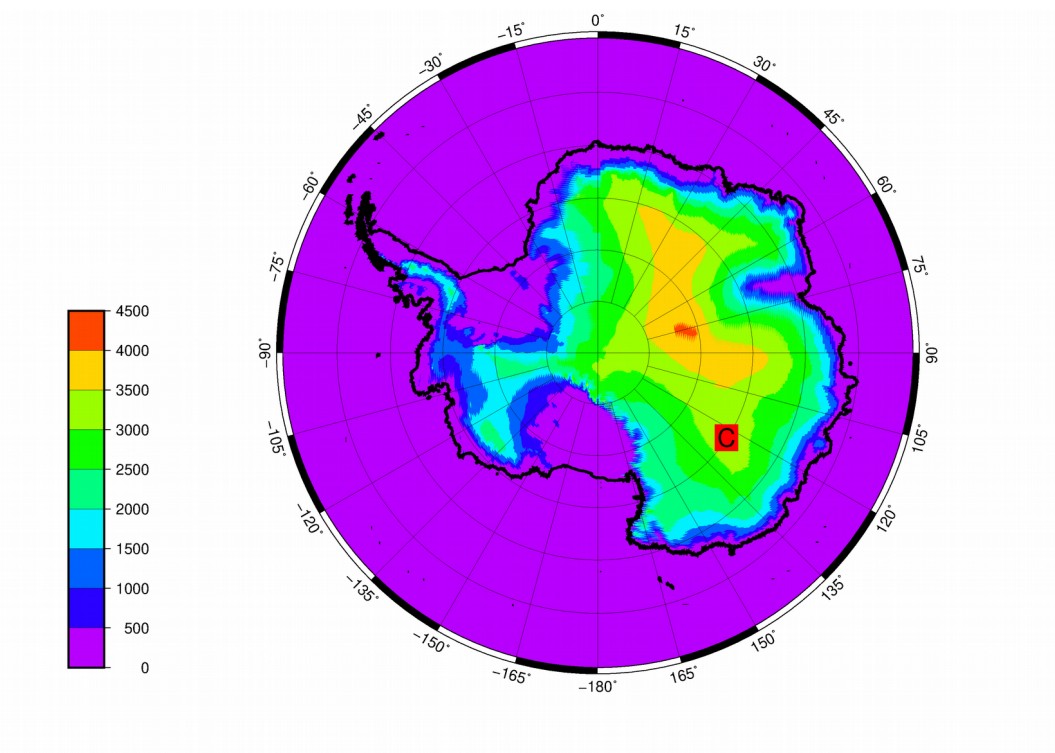

*Figure 1: Antarctic map showing topography (color scale, in m) and position of Dome C / Concordia station (C).*

In 2008, 6 Vaisala HMP45C thermohygrometers were deployed to measure atmospheric temperature and humidity, along with 6 collocated Young 05106 aerovanes for measurement of wind speed and direction. Instrument models were selected

initially for consistency with an observation system deployed at the coast of Adélie Land two years earlier (Genthon et al. 2007). However, this choice proved less than ideal for Dome C, as the HMP45C does not operate below -40°C, and the Young 05106 is a marine oriented aerovane with coated bearings that performed poorly in the extreme cold of Dome C winter. Field testing of other instruments in subsequent years led to the selection of the more recent Vaisala HMP155 thermohygrometer for temperature and humidity measurements, and the Young 05103 aerovane for wind speed and

direction. The 05103 is factory validated to -40°C but cold room tests and experience in the field demonstrate reasonable operation at much lower temperatures, including those encountered at Dome C. The main problem with this anemometer is frost deposition, which impairs propeller rotation, sometimes causing it to stop entirely, leading to underestimates of wind speed and missing data. Regular manual defrosting is necessary by the science support staff at Concordia Station.



The HMP155 is factory validated to -80°C for temperature, adequate for Dome C, using a PT100 (100 Ω) platinum resistance thermometer. Moisture measurements, using a Humicap (©) capacitive sensor, are not reported in the present paper, as the surface atmosphere is often supersaturated at Dome C creating challenging observational conditions. Although supersaturation occurs at high altitudes in the troposphere, it occurs less frequently near the surface. It took several years after the original instrumentation deployment to recognize, and then prove, that supersaturation frequently occurs in the

surface atmosphere at Dome C, and then consequently develop instruments able to observe supersaturation reliably (Genthon et al. 2017). Such observations became operational in 2016. Another issue connected to the particular local environment relates to shielding. At first, passively (wind) ventilated shields were used to protect the thermohygrometers from solar heating. A standard multiplate (gill-style) radiation shield, a type widely employed in the meteorological measurement community including in Antarctica, was used. However, it quickly became apparent that this shield type was inappropriate

on the Antarctic plateau where the wind (section 3) does not consistently blow at sufficient speeds to ventilate the interior of the shield adequately, and the gills do not efficiently protect against strong upwelling solar radiation reflected by the high-albedo surface. The passively ventilated shields were exchanged for mechanically (electric fan) aspirated shields (Young 43502) during the 2009-10 summer season, which required a slight repositioning of the sensor at level 2. This relocation did not significantly change the observations made by this sensor. Since 2010 for temperature and wind measurement, and 2016

for moisture measurement, the instrument types, locations and measurement techniques have remained unchanged. The shorter observational record and special conditions for atmospheric moisture at Dome C are motivation to leave the presentation of humidity for a forthcoming, dedicated paper. This article will therefore focus on wind and temperature. The mean instrument heights above the surface, rounded to account for accumulation over the period of interest, are 3, 10, 18, 25, 33 and 42 m.


All instruments on the tower are sampled at 30-second intervals. Averages, minima, maxima and variances are calculated over 30-minute periods and stored using a Campbell CR3000 datalogger. For wind, the instantaneous U (East-West) and V (South-North) components are calculated from the aerovane wind speed and direction observations, before the modulus U and V components are processed and stored. Yet, because averaging wind direction can be ambiguous, instantaneous samples

are also saved at 1-minute intervals starting in 2015. The resulting dataset over the period 1 January 2010 to 31 December 2019 (2015 to 2019 for 1-minute wind samples) is presented here. The time-series is not fully continuous though, as both instruments and dataloggers stop for servicing and occasionally fail. For example, interruptions occur each year in summer for system maintenance, although they are kept as short as possible. Blackouts also occur sporadically at the station. The instruments most likely to naturally fail are the aerovanes where the moving parts are affected by frost deposition and timely

manual defrosting cannot always occur. The aspirated shields for the thermohygrometers also have moving parts, which are affected by both the extreme cold and frost deposition.



Beyond missing data, detection of instrumental or datalogging failures is not always obvious. The vertical wind speed gradient is generally positive but the occurrence of a low level inertial jet (Gallée et al. 2015) occasionally inverts the

gradient such that the sign of the vertical gradient is not an unambiguous quality test. The temperature gradient is a more reliable quality metric as it may only slightly decrease with height over such a shallow depth. Temperature cannot be above freezing (0°C) at Dome C, and cannot reach below -90°C. Wind speed cannot be less than 0 m.s$^{-1}$ (in fact the manufacturer-stated starting threshold velocity for the anemometers is 1 m.s$^{-1}$ and data below this value should be used with caution), and it does not reach above 30 m.s$^{-1}$ at this location. Data outside of those ranges, or showing suspicious vertical variations or

unrealistically steep changes, are eliminated. Finally, to simplify processing, and for consistency in the vertical structure, only time steps for which valid data are available at all 6 levels are retained. Over the 10-year period presented here, slightly more than 2% of the temperature data is missing, while the missing fraction reaches 22% for wind, highlighting the greater difficulty in measuring this variable in the extreme environment of the Antarctic plateau.

Two AWS also observe the local surface meteorology at Dome C. These stations use different radiation shields for the temperature sensors, none being mechanically ventilated. Therefore, the AWS temperature records may be prone to a radiation warm bias larger than in the tower data reported here (Genthon et al. 2011). The Dome C AMRC AWS is one of the longest standing AWS in Antarctica with station data available since 1984. However, when airborne and satellite surveys of the local topography became available in the 1990s, it was found that the AMRC AWS had been located ~50 km away from

and about 30 m below the very top of the dome. In early 1996, the station moved to the geographical summit. With this relocation, the station name changed in the archive from Dome C to Dome C II. Since then, the station configuration remained stable, particularly over the period 2010-2019 of interest here, with the exception of occasional raising to compensate for snow accumulation. However, station raising is not annual, the instrument height above the snow surface has been variable, and is not recorded in the available archives. The elevation of the Dome C II anemometer (Bendix/Belfort

aerovane) and thermometer (Weed 2-wire bridge PRT) was measured at 245 and 240 cm, respectively, during the austral summer 2016-17. The thermometer is shielded from direct solar radiation by a mere vertical piece of aluminum pipe. This was shown by Genthon et al. (2011) to poorly protect against radiation heating when the wind is less than ~5 m.s$^{-1}$. A second AWS is operating since 2005, deployed as part of the Italian Antarctic program (Grigioni et al., 2019). This station, referred to as AWSIT here, reports temperature at 2 m and wind at 3 m aloft, using a Vaisala HMP45D thermohygrometer, a WAA151

cup anemometer and a WAV151 wind vane.

Simple visual inspection of the tower data identifies the main modes of variability as the seasonal and diurnal cycles. However, quantifying those modes and extracting information on less obvious variability modes requires more objective data analysis methods. Here, harmonic analysis is performed using the correlogram method (Blackman and Tukey, 1958) as

described in Ghil et al. (2002), with data series tapering using a Bartlett window.





## 3. CALVA tower data

### 3-a. Temperature


Fig. 2 displays the evolution of the daily-mean temperature across the ~42-m surface atmospheric column from 2010 to 2019 using the tower thermohygrometer observations, removing sub-daily variability including the diurnal cycle. As shown in the figure, seasonal variability is large, particularly near the surface due to the steep surface-based temperature inversions that develop during the polar night, when the surface radiates thermal energy faster than the atmosphere above due to the larger

emissivity of snow compared to that of the dry atmosphere. Synoptic variability is also large, particularly at the surface in winter due to the strong modulation of the steep temperature inversion by the synoptic disturbances and particularly warm maritime intrusions from the coastal regions (Genthon et al. 2013). Combining diurnal, seasonal and synoptic variability, the temperature can reach from below -80°C (22 June 2017) to as warm as -17°C (2 January 2014), the extremes recorded in the 2010–2019 period. The corresponding vertical profiles along the tower for each of these days are shown in fig. 3. In the

warm case (fig. 3a), the temperature is uniform along the tower to the extent of measurement accuracy. This is a case of summer convective mixing (Argentini et al. (2005), Genthon et al. (2010)). The coldest temperature occurs in winter (fig. 3b), when solar radiation is null and strong surface-based inversions develop in calm conditions. Periods of extreme cold temperatures are generally associated with the strongest inversions and the minimum temperatures observed at the surface. Strong temperature inversions within the tower height generally build-up when non-linear turbulent diffusion vertically

transports cold air from the surface, e.g. when the stable boundary layer transits from a very stable (with a very strong near surface-based inversion) to a weakly stable regime (Vignon et al. 2017). Such inversions can be also amplified by the heating of the air associated with the climatological large-scale subsidence over the dome-shaped Antarctic Plateau (Vignon et al. (2018), Baas et al. (2018)).  The largest daily-mean temperature inversion across the full tower height on record occurs on 24 June 2017 (fig. 3c), 2 days after the occurrence of the coldest surface temperature.




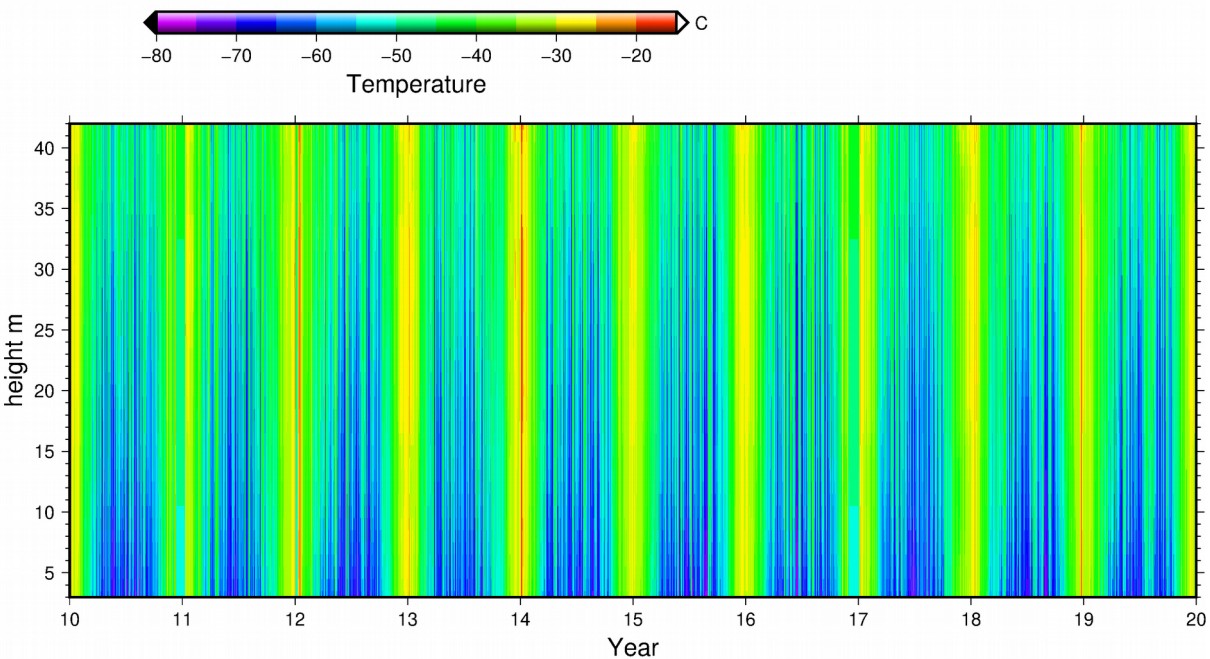

*Figure 2: 10-year record (2010-2019) of daily-mean atmospheric temperature in °C, from the 6 thermohygrometers installed between 3 and 42 m above the surface.*


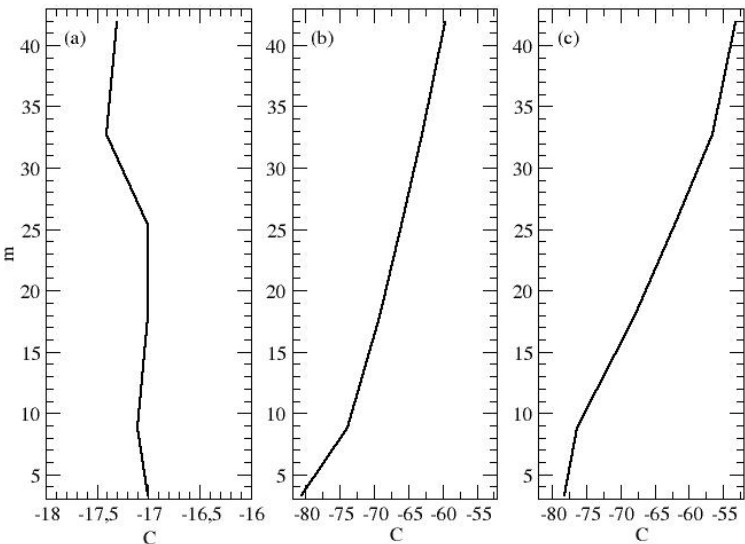



*Figure 3: Vertical profile of daily-mean temperature on 2 January 2014 (a), 22 June 2017 (b) and 24 June 2017 (c), the days with the warmest and coldest temperatures and the steepest temperature inversion, respectively.*


The mean monthly climatology of daily means is summarized in Appendix 1. Concerning the annual cycle, Fig. 4 shows the monthly averages of the daily means over the 10-year period. The interannual standard deviation (not shown) varies between 1.1 and 2.8°C, with the largest variation in winter and no clear dependence on elevation above the surface. The mean monthly temperature is warmest farthest from the surface for all months. The mean temperature gradient is small in summer,

but it increases in winter up to an average of 0.4°C.m⁻¹ along the tower in June. It has long been noticed that the annual cycle of temperature on the Antarctic plateau, rather than quasi-sinusoidal, has short summers and long flat winters, known as the "coreless winters" (Wexler, 1958), with sharp transitions in between. Fig. 4 shows that the "coreless winter" is increasingly flatter as height above the surface increases. This is because as the inversion increases (decreases) in fall (spring), the surface radiative cooling (warming) is increasingly less (more) propagated to the air layers above by turbulence. Fig. 5 shows the

temperature difference between that observed at the lowest tower level and those observed by the 2 AWS. The tower and AWSIT temperature measurements agree well in December and January, when turbulence and/or convective mixing ensures that differences in instrument height above the surface have a limited impact. In winter, even small elevation differences can induce significant air temperature deviations. The 1st tower level is 1 to 2 m higher than the ~2-m AWSIT level, which itself has small annual variations in height above the surface due to snow accumulation. This may well account for the 1–2 °C

temperature difference between datasets in the winter. The same reasoning applies to the AMRC AWS temperature in winter. On the other hand, the large (up to 4°C on average) warm bias of the DOME C II dataset in summer compared to the tower data is the signature of poor shielding of the AMRC AWS temperature sensor to solar radiation (Genthon et al. 2011). It is assumed that the two AWS stations experienced the same accumulation as the tower and so the relative height difference among the datasets remained consistent throughout this period.




Earth System Discussions
Science
Data
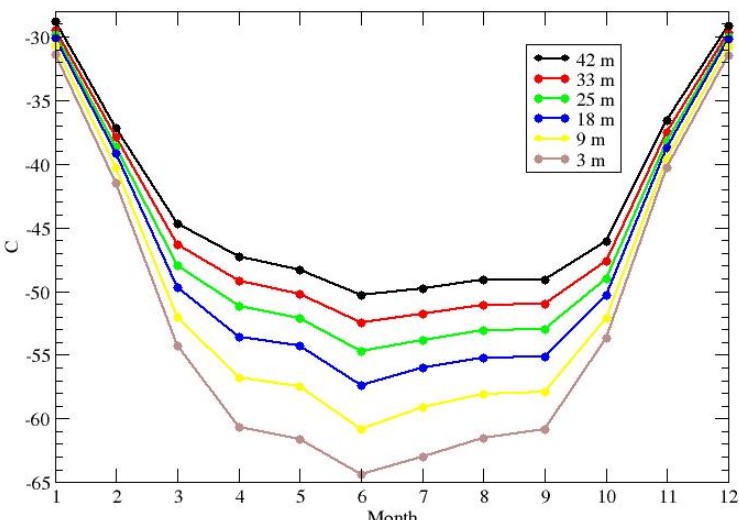

Figure 4: Mean 10-year seasonal cycle of temperature at the 6 Dome C tower levels. Elevation (legend) is rounded since it changed by almost 0.8 m in the course of the 10-year period.


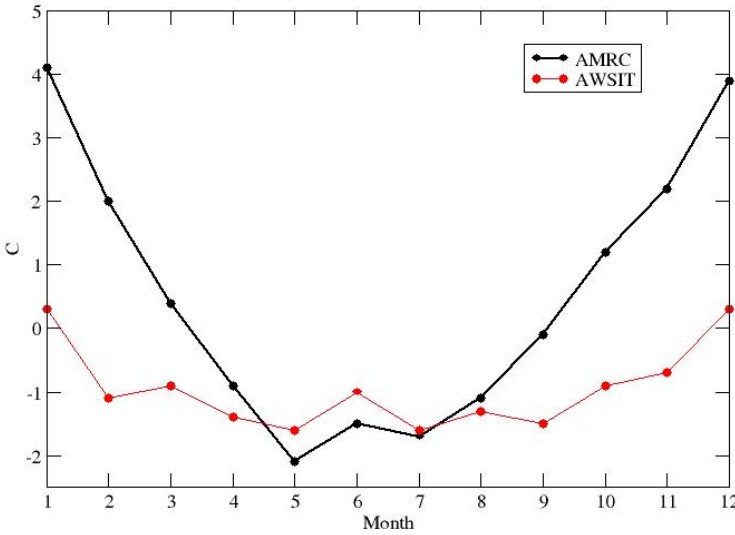

Figure 5: Temperature difference between the lowest temperature sensor on the Dome C tower (~3m) and the nearby stations AMRC AWS (black) and AWSIT (red) (~2m).





Fig. 6 shows the results of the harmonic analysis of the temperature data in the frequency ranges with the largest spectral power peaks. Both the diurnal (fig. 6a) and the annual/semi-annual (fig. 6b) cycles are most pronounced near the surface; the amplitude decreases with elevation above the surface. At diurnal time scales, the cycle almost fully vanishes at the top of the tower. This is consistent with, and is further illustrated by, the 4-day samples shown in Genthon et al. (2013) (their fig. 7). Vertical dampening occurs over a much shallower layer near the surface at diurnal time scales than at annual time scales.

Besides diurnal and annual cycles, the only significant cycle found is semi-annual. The semi-annual oscillation in the mid and high southern latitudes consists of the twice-yearly contraction and expansion of the low pressure belt around Antarctica, in response to differences in heat storage between Antarctica and the surrounding oceans (van Loon, 1966). As a result, various climate variables such as surface pressure, winds and temperature in the middle and high latitudes show a half-yearly wave (van den Broeke et al. 1998). This semi-annual signal is reflected in the full depth of the boundary layer at Dome C

(fig. 6b). The amplitude of the temperature cycle varies by a factor of less than 2 along the tower, while the gradient is much larger at annual and, particularly, at diurnal time scales. This reflects the fact that the diurnal and the annual cycle are controlled by the surface energy balance. The energy balance is largely modulated by the cycles of local solar radiation input, while the semi-annual cycle results from large-scale processes, with the bulk of the atmospheric column impacted, in turn influencing the boundary layer from above.


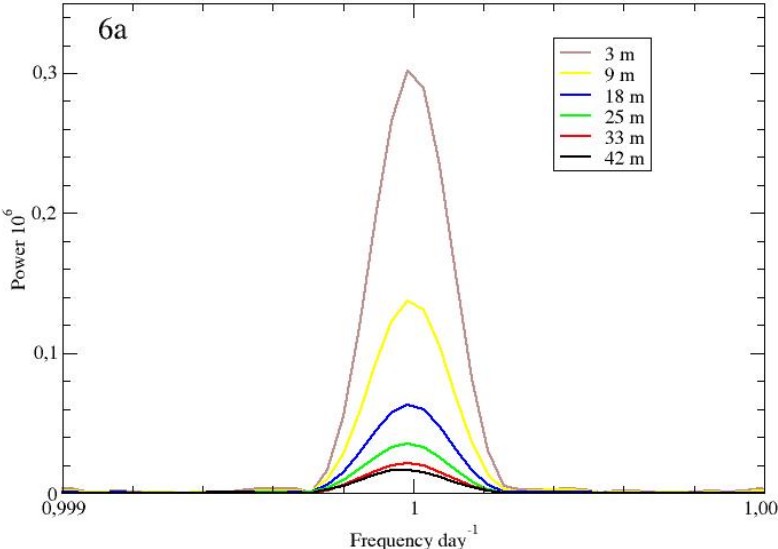





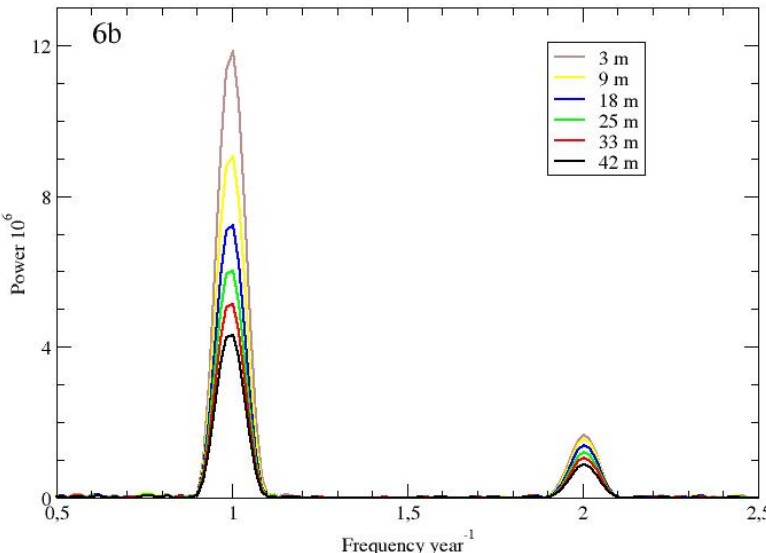

*Figure 6: Harmonic analysis of the temperature time series near diurnal (6a) and annual/semi-annual (6b) frequencies.*

Fig. 7 displays the evolution of the annual-mean temperature over the decade 2010-2019 at the 6 levels above the surface. The interannual variability is similar at all levels, albeit with larger amplitude at lower levels, with the warmest temperature in 2011 and the coldest in 2016 at all heights on the tower. This variability is thus unlikely related to sensor defects at the particular level. There is a decreasing temperature trend at all levels. The linear regression slope ranges between -0.08 and -0.17°C per year, with the smallest (largest) trend at the upper (lower) level. However, considering the small sample size and

the large interannual variability, the linear trends have limited statistical significance. T-testing indicates that only the trends at levels 4, 5 and 6 (25, 33, and 42 m in fig. 4) above the surface are significantly different from 0 at the 95% confidence level. In addition, part of the trend is because as snow accumulates the sensors get closer to the surface and thus sample colder air layers in the surface-based inversion. The mean temperature gradient ranges from 0.16°C m$^{-1}$ at the top (between levels 5 and 6) to 0.47°C m$^{-1}$ at the foot (between levels 1 and 2) of the tower. Just considering the mean vertical gradients,

an 80-cm (8 cm per year over 10 years) lowering of the sensors results in 0.13 to 0.38°C of apparent cooling at the highest (42 m) and lowest level (3 m), respectively.

Although trends have been reported from other longer time series (Ricaud et al., 2020b) and partly explained by changes in the Southern Annual Mode (Turner et al. 2019), here the short series, weak significance and large relative impact of change

in elevation prevent any firm conclusion about the ambient temperature trend in the lower atmospheric boundary layer from these data. Seasonal trends (not shown) actually suggest slight warming in summer (Dec-Jan) and winter (Jun-Jul-Aug) but with an even lesser number of samples and more variability this is an even less reliable result.



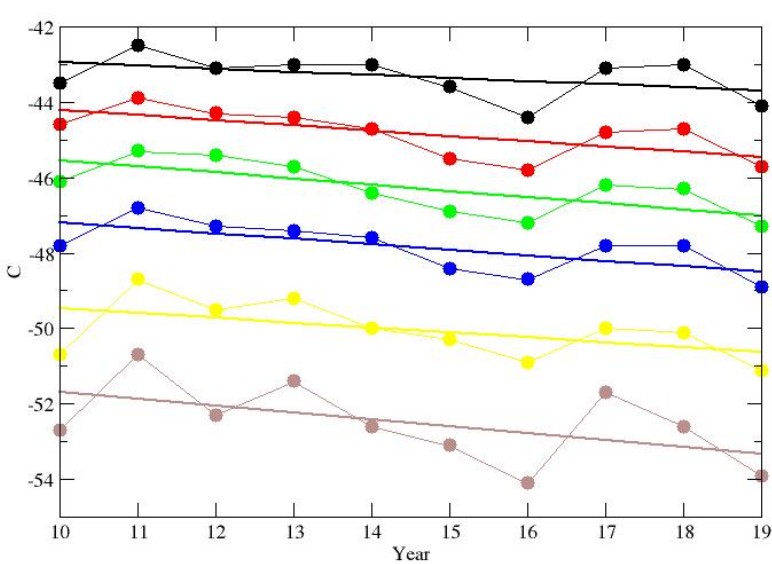

*Figure 7: Annually averaged temperature over the period 2010-2019. Coldest temperatures occur at the lowest model level.*

**285   3-b Wind speed and direction**

The average daily-mean wind speed across the 42-m surface atmospheric layer sampled by the tower instrumentation is shown in fig. 8 from 2010 to 2019. It is important to recall that in spite of data screening, the reports might underestimate the true wind speed due to frost deposition and occasional impediment to proper operation, particularly at low wind speed.

Furthermore, the manufacturer estimates the instrumental wind speed threshold at 1 ms$^{-1}$ such that even in more conventional conditions, this sensor would underestimate the contribution of weak winds cases. Averaging the wind speed daily removes sub-daily variability including the diurnal cycle. Major periods during which the data are consistently missing or discarded due to quality control criteria are blacked out. In comparison with AWSIT wind speed records over the same period and time steps, the difference with the lowest tower level is larger than 2 m.s$^{-1}$ (AWSIT showing stronger wind) 3.5% of the time, and

smaller by the same amount 1.2% of the time. Considering that wind speed measurements are particularly prone to errors in this extreme environment, this is a reasonable correspondence. However, although the coldest temperatures are found near the surface (section 3a), local staff often report that frost deposition is more abundant at the higher levels. For wind measurements, the aerovane performance near the surface may not be representative of instrument performance all along the

tower. Indeed, fig. 8 indicates extensive periods in the record where wind observations failed the quality control process for

at least one level on the tower.

The temperature variability is largest near the surface, which is the signature of the strong influence of the ABL inversion that modulates the free atmosphere forcing of the near-surface atmosphere. However, the wind is less directly affected by solar radiation; diurnal and seasonal wind variability near the surface are much weaker than for temperature. The site is not

locally subject to katabatic winds, which blow over much of Antarctica, because the local surface slope is very small. Here, air momentum is essentially of synoptic origin in the free troposphere, propagating down to the surface through boundary layer mixing. Thermal stability dampens turbulent mixing and thus propagation of free atmosphere momentum to the near surface. Weak dynamic coupling between the surface and the free atmosphere favors strong wind shear in the ABL, such as in cases with almost no wind (1 m.s$^{-1}$ or less) 3 m above the surface while reaching near 15 m.s$^{-1}$ 40 m above.


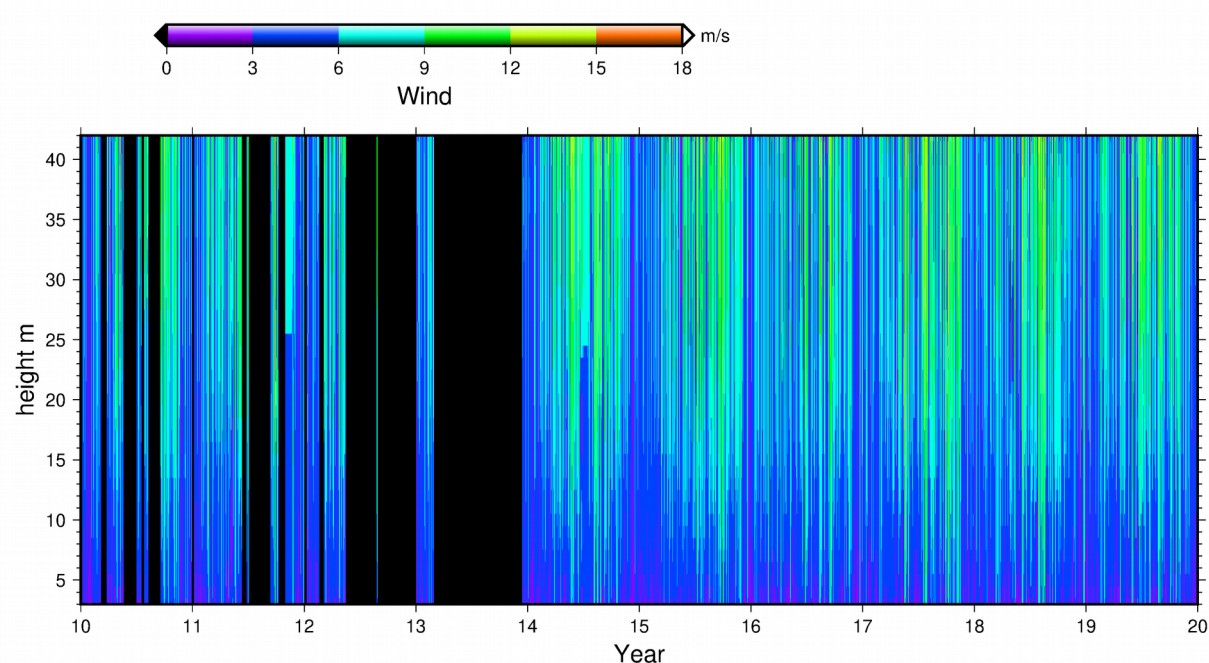

Figure 8: Ten-year evolution of daily-mean wind speed in m s$^{-1}$, between 3 and 42 m above the surface, from 2010 to 2019. Black shading shows major periods for which data are discarded during quality control or missing (see section 2).






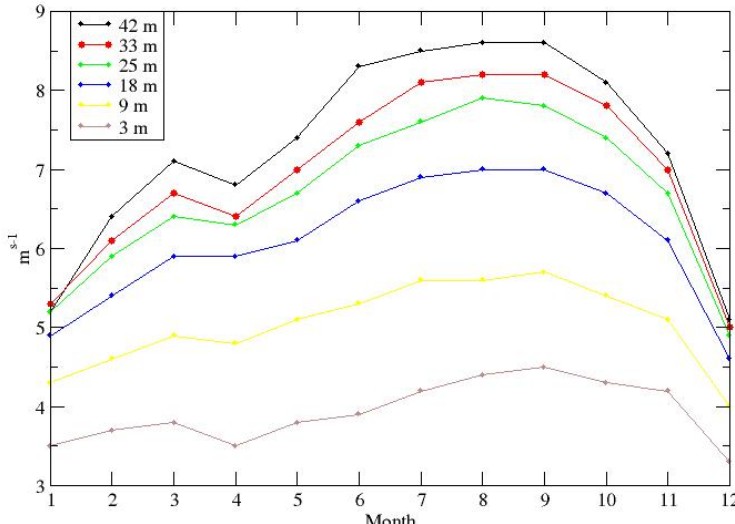

*Figure 9: Mean 6-year seasonal cycle of wind speed at the 6 levels along the tower. Years 2010 to 2013 are not included in the averaging process due to large gaps in the data in these years (fig. 8).*


While the seasonal cycle is visually straightforward for temperature (fig. 2), which directly responds to the local seasonal cycle of irradiation, it is less obvious for wind (fig. 8). This is better illustrated by fig. 9, which shows the annual cycle of monthly-mean wind speed at the 6 levels on the tower averaged over 2014-2019. The maximum wind speed occurs in local late winter and early spring due to synoptic forcing. This 3-month period is when the temperature gradient across the mid to

high southern latitudes is largest due to differential insolation. For example, in August it is still polar night at 75°S while at 40°S, the solar input at the top of the atmosphere has already increased by 40% since austral winter solstice, (Peixoto and Oort (1991), their fig. 6.4). The relative amplitude of variability at the various heights above the surface can be illustrated using harmonic analysis. This is shown in fig. 10 for the near daily (fig. 10a) and annual / semiannual (fig. 10b) periods, the only time ranges with significant spectral power peaks.


The vertical distribution of spectral power is almost opposite to that of temperature (fig. 6), further illustrating that forcing is at the top of the air layer for the wind, while it is at the surface for temperature. This is not the case for the 2 levels closest to the surface though; there is more power at 9 than at 3 m. This is consistent with the "crossing point" concept introduced by van de Wiel et al. (2012). In summer, assuming a constant geostrophic wind, when the inversion builds at "night" the

transport of momentum toward the surface by convection stops. The wind near the surface decreases. Above the surface, on the other hand, the wind accelerates due to the development of the night time inertial jet (Gallée et al., 2015). There is a height at which the wind is relatively constant throughout the day, a "crossing point" where spectral power is minimal at diurnal time scale. At Dome C, this was estimated at about 10m by Vignon et al. (2017a). This is precisely the elevation at which the harmonic analysis shows minimum power.




Another difference of the wind relative to the temperature is that the diurnal and semi-annual cycles are much larger relative to the annual cycle for the wind. Again, the semiannual cycle for the wind speed is mostly explained by the large-scale dynamics while for temperature, it is primarily a response to local solar forcing. The fact that the semiannual power peak for wind (fig. 10b) is slightly shifted to periods shorter than a half-year probably reflects that wind data is noisy. As mentioned

above, wind speed is relatively difficult to measure and only 6 years of data can be confidently retained for this analysis.

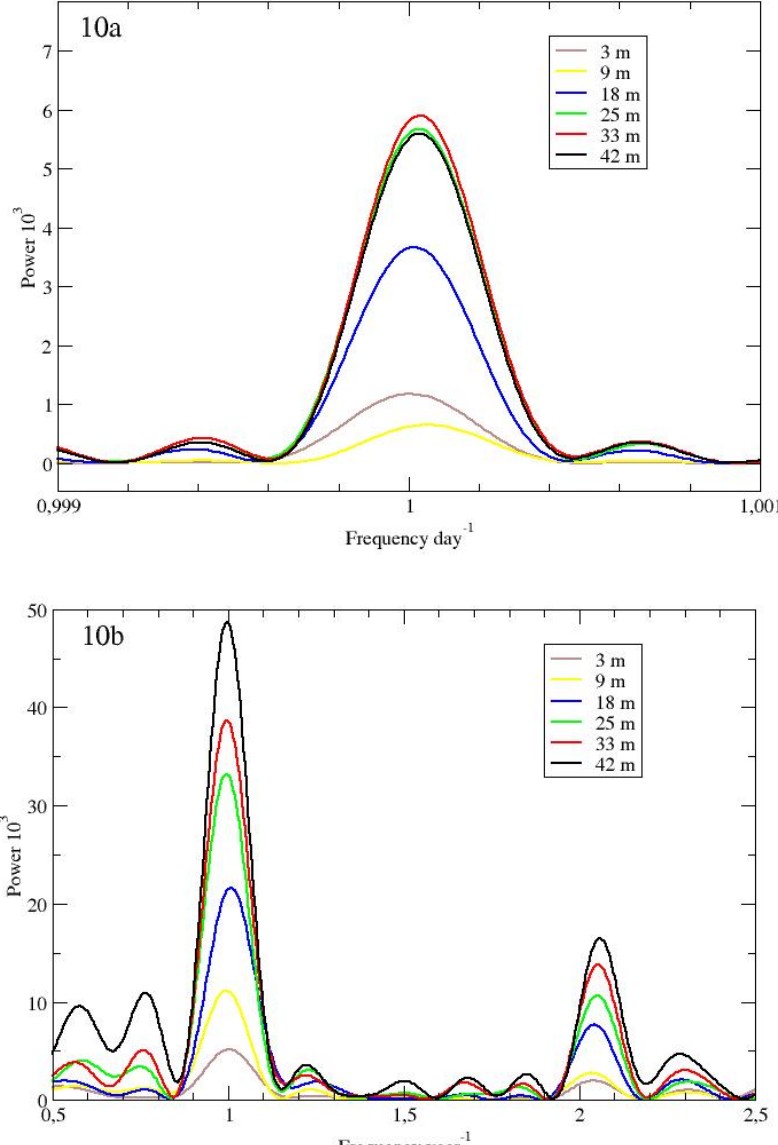



*Figure 10: Harmonic analysis of the 2014-2019 wind speed times series near diurnal (10a) and annual / semi annual (10b)*
*frequencies.*

Finally, fig. 11 displays the probability distributions of wind direction and corresponding wind speed at the various levels along the tower. This is calculated from the 1-minute instantaneous wind observations (see sampling discussion in section 2) binned in 20° longitude intervals; this may be compared with the wind rose reported by Aristidi et al. (2005, their fig. 3)
from the AMRC AWS Dome C II data. The results are broadly consistent, both showing a favoured wind direction in the vicinity of 180°. In Antarctica, the surface wind directional constancy is generally very high due to the katabatic wind regime that is controlled by the surface slope (Parish et al. 2003). This is not the case at Dome C because the slope is locally null. Thus, a predominant wind direction results from a large-scale, synoptic control.

The probability distribution of the wind direction data in the bottom two levels suggests the occurrence of wind turning in the shallow Dome C ABL (Rysman et al. (2016) , fig. 11). Genthon et al. (2010) also report evidence of wind turning across the boundary layer in summer when the sun is lowest above the horizon, then vanishing as the temperature inversion is broken by convection when the sun rises up in the morning. Their results employed 30-minute averaged wind directions obtained from 30-minute averaged U and V wind components.


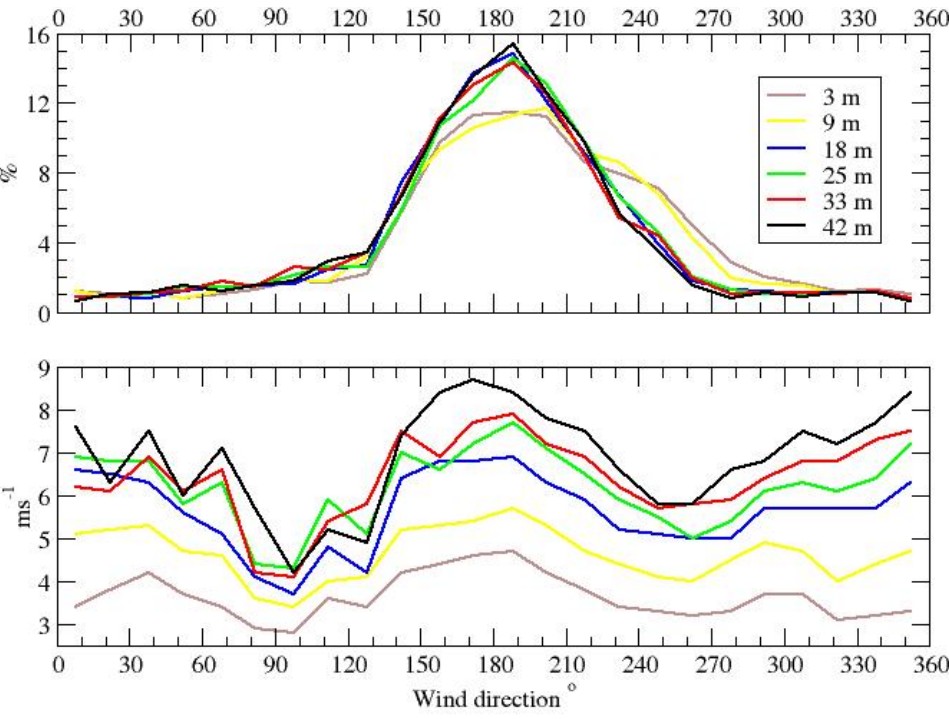



*Figure 11: Probability distribution (upper plot) and mean wind speed (lower plot) according to wind direction.*

## 4. ERA5 and the Dome C ABL

Because the Dome C ABL properties are extreme and differ significantly from that in other regions, even the most up-to-date atmospheric models with a large community of users and a large range of geographical and topical applications may have large deficiencies in this region, even with respect to 1[st] order aspects. To illustrate this, ERA5 reanalyses of temperature are compared with the tower observations. Correlations integrate bias with both the amplitude and timing of variations at all time scales. Harmonic analyses allow for a comparison at specific periods of largest variability from diurnal to annual time

scales.

Fig. 12 displays the correlation between the tower-observed (abscissa) and ERA5-analyzed (ordinate) temperature in the near-surface ABL at Dome C for 2010-2019. ERA5 analyses are available at a 1-hour time step. For each reanalysis time step, the temperature from the tower profile at native ½-hour time step is linearly interpolated to the two lowest model levels.

A logarithmic, rather than linear, vertical temperature change is expected in a stable temperature profile. However, the difference in elevation between corresponding tower and model levels is only a couple of meters at most and a linear interpolation is a reasonable approximation here. The elevation of the levels in the model varies in time because the model uses a hybrid sigma vertical coordinate. Over the analysis period, the elevation of the first level fluctuates between 6.9 and 8.9 m (7.8 m on average), while the second level varies between 21.6 and 27.4 m (24.1 m on average). The elevation of the

third model level varies between 37.5 and 47.3 m and is above the top of the tower most the time. Since the most pronounced aspects of the boundary layer inversion occur below that level, no extrapolation is attempted for comparison at this level. On the figure, the black line indicates the 1:1 bisector.

The reanalysis product is generally colder than the observations in the summer. The mean temperature bias is similar at the

two levels (2.5 and 2.3°C at the higher and lower levels respectively). In winter, the mean bias is larger than in summer at the lower level (2.6°C) but smaller at the higher level (0.6°C). However, in winter the reanalysis product is cooler at warmer temperatures and warmer at very cold temperatures. In fact, the reanalysis never produces the very cold temperatures observed at Dome C in winter. The correlation coefficient (reported on fig. 12) is larger in summer (~0.85) than in winter (0.72 or 0.73), reflecting a large contribution of diurnal variability by the solar cycle in summer. For the same reason, the

standard deviation agrees better in summer (3.7 vs 3.7 °C (ERA5 vs tower, upper level) and 4.0 vs 4.4 °C (ERA5 vs tower, lower level) than in winter (8.8 vs 6.6 °C and 8.4 vs 6.2 °C).



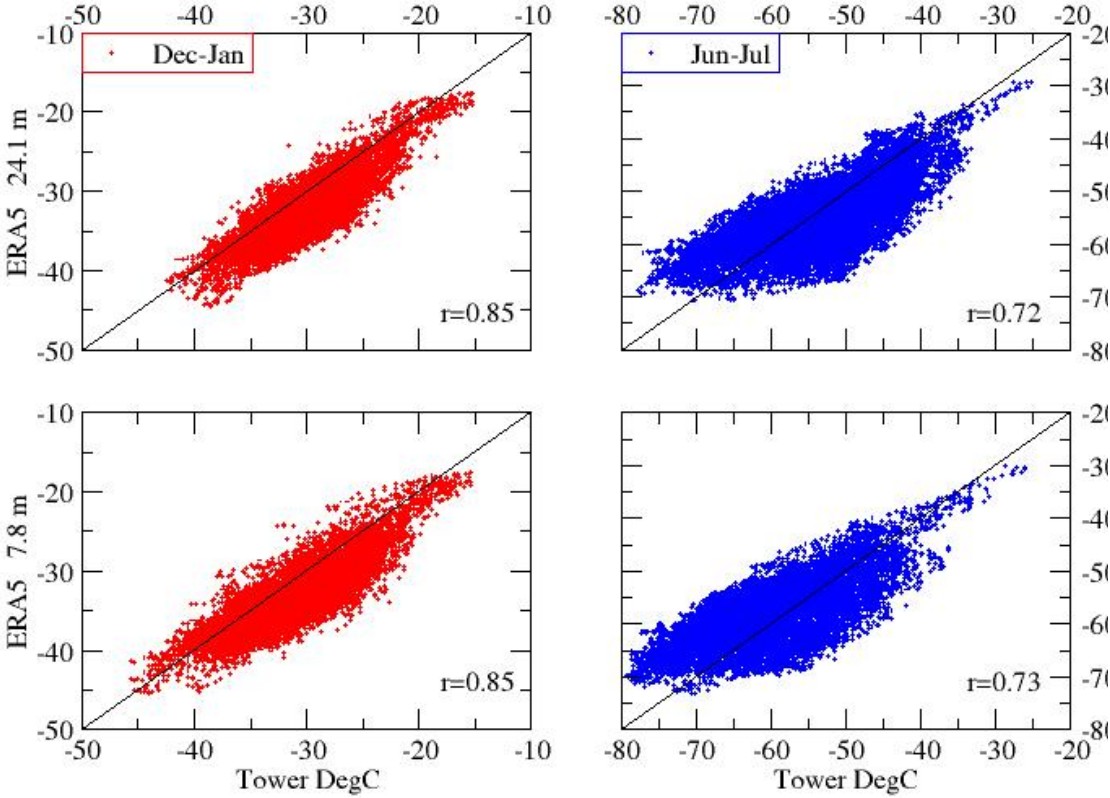

*Figure 12: Scatter plots of ERA5 versus observed tower temperature on (in °C) at the 2 lowest ECMWF model levels above the surface, 7.8-m (lower plots) and 24.1-m (upper plots) for summer (red) and winter (blue) conditions. Tower temperature*
*is linearly interpolated on to the model levels. The black line is the 1:1 line.*




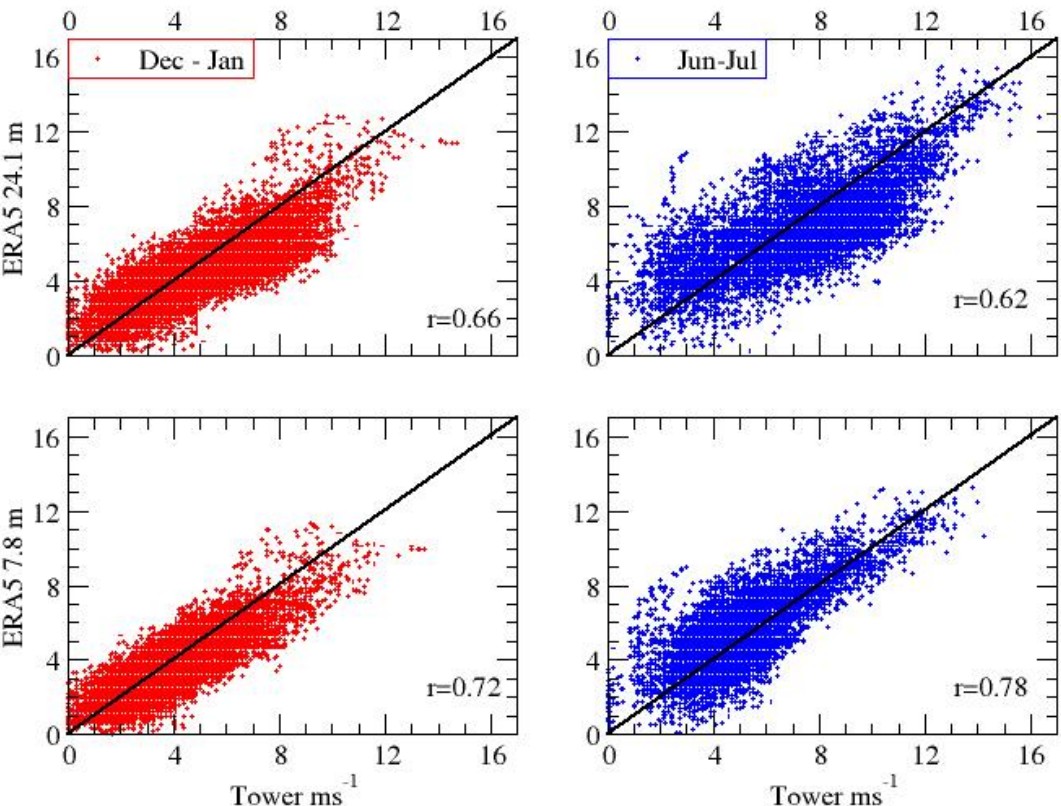

*Figure 13: Scatter plots of ERA5 reanalysis wind speed versus observed tower wind speed (in m.s$^{-1}$) at the 2 lowest ECMWF model levels above the surface, 7.8-m (lower plots) and 24.1-m (upper plots) above surface for summer (left-hand column) and winter (right-hand column) conditions.*

Concerning the wind, fig. 13 shows the correlation between tower-observed (abscissa) and ERA5-analyzed (ordinate) data. Reanalyzed wind speeds agree generally with the tower-observed wind speeds, although there can be significant differences of greater than 2 m.s$^{-1}$. The correlations (0.66 and 0.72 at the upper and lower levels) is less than for temperature in summer, reflecting a lesser "pacemaker" control by solar forcing for the wind. The correlations are more of a similar order in winter when variability is of synoptic and thus more stochastic origin, the correlation being even larger for the wind at the lower level (0.78 for wind vs 0.73 for temperature). It appears that in the lowest model layer there is a tendency to slightly underestimate the wind speed relative to observations in the summer and overestimate in winter, however this may just be an artifact of interpolation or instrument error. In the upper model level, the summertime underestimate of wind speed remains, while in winter, it appears that the reanalysis is lower than the observations at wind speeds above 6 m.s$^{-1}$ and higher below.



The standard deviation ranges from 1.5 to 2.3 m.s$^{-1}$ with higher values in winter and in the higher level, and is mostly underestimated by the model, except in winter at the lower level where it agrees well.

From harmonic analysis, Fig. 14 compares the amplitude of the diurnal, semi-annual and annual cycles of temperature in ERA5 reanalysis and in the observations. The first 3 ERA5 levels are shown, including a 3$^{rd}$ level above the top of tower but close enough in the present case to compare how the amplitude of the cycles vary with elevation. The three closest levels are shown for the observations. The reanalyses reproduce a decreasing amplitude with elevation, as seen in the observations although less pronounced. On the other hand, the reanalyses consistently underestimate power at the frequency scales shown.

There are very few observations available in the boundary layer to control the production of the reanalyses, compared to the free atmosphere where space borne sounders, in particular, provide essential data. At Dome C, there is only one radiosonde launch per day delivering data to the global telecommunication system. The analyses in the boundary layer are thus largely model data and therefore reflect boundary layer model limitations. The underestimation in temperature is particularly marked at the diurnal time scale because variability is largely controlled by the boundary layer processes and their ability to build an

inversion when the summer sun is lower on the horizon. The impact of this limitation emerges at all time scales (Fig 14c).

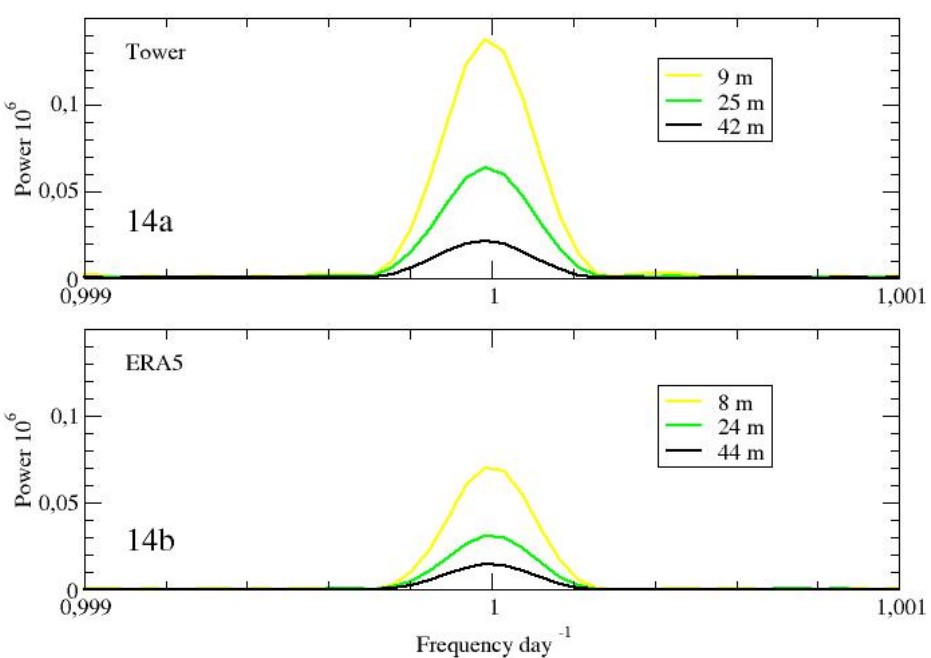



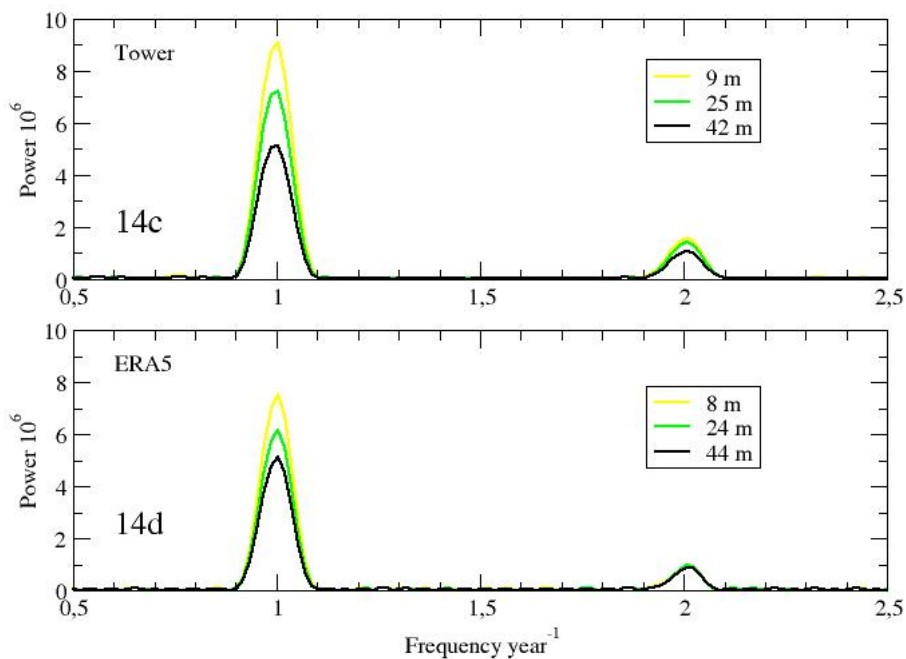

*Figure 14: Harmonic analysis of the temperature times series near diurnal (14a, 14b)) and annual / semi annual (14c, 14d)) frequencies.*1.2 Subsection

**5. Discussion and conclusion**

The installation of a 30-m meteorological tower at Dome C in 2004 provided an excellent opportunity to study the Antarctic atmospheric boundary layer. In 2008, the tower was raised an additional 15-m, and equipped with a suite of instrumentation distributed at approximately regular intervals along the tower. The additional height provided a more complete profile of the near-surface atmospheric properties, especially in winter when the ABL is quite thin (Genthon et al. 2013). The establishment of a long time-series (10+ years) of meteorological properties has led to significant advances in the observation and understanding of the ABL properties over the Eastern Antarctic Glacial Plateau.

The harsh environment at Dome C created unique observational challenges, which required adaptation of standard polar observing techniques and, in some cases, the development of novel approaches. For example, the accumulation of thick frost layers on any structure in all seasons, but especially during the long polar winter, can impede aerovane motion and block thermohygrometer ventilation. Currently, this requires the regular intervention of station scientific staff. Also, the intense solar radiation, combined with frequent low wind speeds in austral summer, can cause the thermohygrometers to overheat in passively ventilated sensor shields. This can be overcome with mechanically ventilated shields, but increases daily and



seasonal maintenance due to frost build-up and wear on the internal fan. Despite the observational challenges, these unique data have provided important insights into the dynamics of the near-surface atmospheric boundary layer. Previous studies with the tower data have established the occurrence of convection in austral summer during periods of high solar elevation

angle (Genthon et al. 2010), extreme surface-based temperature inversions (Genthon et al. 2013), stable boundary layers in both seasons (Vignon et al. 2017a), low-level inertial jets (Gallée et al. 2015). These data have also been used to evaluate model simulations (Ricaud et al. 2020a, Couvreux et al. 2020). The accumulation of 10+ years of data from the 45-m tower allows for the development of a climatology of the near-surface boundary layer, which can be compared with other climatologies established using surface-based remote sensors and radiosondes (Marshall (2002), Ricaud et al. (2020b)) and

reanalyses.

The climatology developed from the CALVA tower data demonstrates the strong influence of insolation on the near-surface temperature, with the largest variations closest to the surface, and steep vertical temperature gradients in the winter. The annual cycle of temperature clearly shows that summer is brief and relatively warm with some vertical mixing, followed by a

sharp transition to a long cold winter, a phenomenon known as the « coreless winter » (Wexler 1958). Harmonic analysis also illustrates that the temperature varies most strongly at diurnal, annual and semi-annual frequencies. Interannual variability in the monthly mean temperature varies from 1.1 - 2.8°C, with the smallest variability occurring in summer. There is some indication of a cooling trend in the annual mean temperature from 2010-2019, but this result is only significant at the 95% interval, for only 3 levels of the tower. The trend in the annual mean temperature of the lower levels is significantly

impacted by the snow accumulation causing a relative « lowering » of the sensors.

The record of wind speeds has more data gaps than the temperature record due to difficulties maintaining continuous instrument operation particularly in the early part of the record. The highest wind speeds occur in late winter to spring, with a secondary maximum in wind speed in autumn and significantly lower wind speeds near the surface. There is less interannual

variation in the wind speed gradient along the tower than in the temperature gradient. The overall distribution of spectral power for the wind speed is similar to that seen in the temperature data with peaks in power at diurnal, semi-annual and annual scales. However, unlike the temperature, the wind speed spectral analysis shows the greatest forcing at the top of the tower, reflecting that the winds are forced by large-scale forcing from above. Wind direction is predominantly from 150-240° (southerly) with a slightly larger range in the bottom two tower levels, resulting from the presence of wind turning in the

shallow Dome C ABL.

Comparison of the lowest tower level with the nearby automatic weather stations indicates that while there is some general agreement among the stations, differences in ventilation techniques and instrument height cause notable disparities among the datasets. Both AWSIT and the AMRC Dome II temperature data show a warm difference relative to the data for lowest

level of the 45-m tower, especially in winter when the temperature gradient is large. When comparing the CALVA tower data



with the two lowest levels of ERA5 data for the same period (2010-2019) relatively good agreement is found between the reanalyzes and observed temperature for the summer when the near-surface atmosphere is relatively well mixed, with a slight bias for cooler reanalyses than observed. However, in the winter, the reanalyses tend to soverestimate the coldest temperatures and underestimate warmer temperatures suggesting a less steep vertical temperature gradient. In addition, the reanalyses appear unable to produce the most extreme cold episodes. There is also reasonable agreement between the reanalyses and observed wind speeds in summer. However, there is less good agreement in the winter, with a notable spread in the observations at low wind speeds, and a slight overestimate in the wind speed in the lowest model layer.

Continued data collection at this important site on the Eastern Antarctic glacial plateau will continue to improve understanding of the near-surface boundary layer processes in this extreme environment.  Not addressed here but left for a forthcoming paper, the enhanced humidity measurements will provide insight into cloud formation (Ricaud et al. 2020a) and precipitation processes that are critical components of the energetic and mass balances. In addition, these tower observations can be combined with radiosondes, and ground-based and satellite remote sensors to produce more complete profiles of the atmospheric boundary layer. This information is essential for improving weather and climate forecasts in polar regions.

**Appendix 1: Ten-year monthly climatology and statistics of daily-mean temperature**

Table 1 displays the statistics of daily-mean temperature in the 2010-2019 period, for each month of the year and each level along the tower. Numbers are rounded to the nearest integer. Averages are shown in black bold, minima and maxima in red and blue respectively. Absolute daily mean maxima are in bold red: They occur in January, similarly at all levels along the tower because of convective mixing. Absolute minimum is in bold blue: this occurs in June at the level closest to the surface associated with steep temperature inversion.

|      | Jan            | Feb            | Mar            | Apr            | May            | Jun            |
|------|----------------|----------------|----------------|----------------|----------------|----------------|
| 42 m | -29 (-17/-37)  | -37 (-28/-50)  | -45 (-33/-65)  | -47 (-34/-68)  | -48 (-29/-68)  | -50 (-33/-67)  |
| 33 m | -29 (-17/-38)  | -38 (-29/-51)  | -46 (-33/-65)  | -49 (-30/-70)  | -50 (-30/-70)  | -53 (-33/-71)  |
| 25 m | -30 (-17/-39)  | -39 (-29/-51)  | -48 (-34/-66)  | -51 (-35/-72)  | -52 (-31/-72)  | -55 (-34/-74)  |
| 18 m | -30 (-17/-39)  | -40 (-29/-52)  | -50 (-34/-68)  | -54 (-37/-74)  | -54 (-32/-73)  | -57 (-35/-75)  |
| 9 m  | -31 (-17/-40)  | -41 (-29/-53)  | -52 (-35/-70)  | -57 (-38/-75)  | -57 (-34/-75)  | -61 (-35/-77)  |
| 3 m  | -31 (-17/-41)  | -42 (-29/-55)  | -54 (-35/-70)  | -61 (-41/-76)  | -62/ (-35/-78) | -64 (-36/-80)  |
|      | Jul            | Aug            | Sep            | Oct            | Nov            | Dec            |





| 42 m | **-50** (-32/-68) | **-49** (-31/-65) | **-49** (-34/-71) | **-46** (-32/-61) | **-36** (-26/-50) | **-29** (-19/-39) |
| 33 m | **-52** (-33/-69) | **-51** (-33/-68) | **-51** (-35/-74) | **-47** (-32/-62) | **-37** (-27/-51) | **-29** (-19/-39) |
| 25 m | **-54** (-34/-71) | **-53** (-35/-74) | **-53** (-36/-76) | **-49** (-34/-63) | **-38** (-27/-52) | **-30** (-19/-39) |
| 18 m | **-56** (-34/-73) | **-55** (-37/-75) | **-55** (-37/-76) | **-50** (-35/-64) | **-38** (-27/-52) | **-30** (-19/-39) |
| 9 m | **-59** (-34/-76) | **-58** (-38/-76) | **-58** (-37/-77) | **-52** (-35/-64) | **-39** (-28/-52) | **-30** (-19/-41) |
| 3 m | **-63** (-34/-78) | **-62** (-38/-77) | **-60** (-37/-77) | **-53** (-33/-67) | **-40** (-29/-53) | **-31** (-19/-41) |


Table 1: Ten-year 2010-19 monthly statistics of daily mean temperature along the 42-m tower.

**Acknowledgments:**

Since 2008, the French Polar Institute IPEV provides logistical support to program CALVA-ACDC (in situ data for CALibration – VAlidation of meteorological and climate models and satellite retrieval, Antarctic Coast to Dome C, IPEV program 1013), thanks to which meteorological observing systems have been deployed and are annually serviced. Some travel for this field work was also supported by the National Science Foundation. The authors cannot list but wish to thank all the people who have contributed in deploying, maintaining and ensuring proper operation of the observing systems on the

field, not the least in winter when one has to climb a 40-m tower with ambient temperature sometimes as low as -70°C, particularly taking account of windchill effect. Authors also thank the French space agency  CNES for support as part of project EECLAT (Expecting EarthCARE, Learning from the A-Train). Thanks to Bas van de Wiel, Nikki Vercauteren, Adam Mohanan and Carsten Abraham in the framework of their respective research projects, Eric Bazile, Olivier Traullé,and leaders and contributors of the GABLS4 project, for feedbacks on the data strength, weaknesses and suggestions for

potential improvements. The AWSIT data were downloaded at https://www.climantartide.it/index.php?lang=en, while Paolo Grigioni at ENEA kindly provided additional details such as instruments used and height above surface. The AMRC AWS data were download from https://amrc.ssec.wisc.edu/data/.

**Data availability:**


The half-hourly data presented here are made available on PANGAEA open data repository (Genthon et al. (2021a, 2021b), https://doi.pangaea.de/10.1594/PANGAEA.932512,   https://doi.pangaea.de/10.1594/PANGAEA.932513).  Processed   data such as daily and monthly means or spectral analysis results, or more voluminous high resolution (minute) data, are freely available on request to the authors




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
