# Peer review of "Temperature and wind observation from 2010 to 2019 on a 45-m tower at Dome C, East Antarctic plateau."

_Earth System Science Data, 2021_

## Author Response (AR1)

**Anonymous reviewer 1:**

Thanks for this paper that describes a 10 year observational dataset from a tall tower instrumented at 6 levels at Dome Concordia Antarctica. Here are the comments, suggestions and recommends I have for this manuscript:

=> We thank the reviewer for insightful and very pertinent comments to improve the paper. Here is how we have taken comments into account and thus hopefully fulfilled the reviewer's requirements. Please note, line numbering is that of the original manuscript, obviously changed in the new manuscript. The new manuscript also accounts for comments by the other reviewer and changes made in response to these comments.

• A minor point, ten years does not quite make a climatology which should be 30 years, but it is understood this all the data you have. (see below for comment on wind data amount...)

=> Yes, we are short of the standard 30-year climatology and although we make parsimonious use of the word "climatology" this is stricto sensus inappropriate. Our series is only 10 years long. We have changes "climatology" where appropriate, into time series or equivalent formulations.

• Line 25 - "See data availability section" is not needed in the abstract.

=> In fact, the abstract should even mention the data link. Citing ESSD"s "User guidance in a data availability section » https://essd.copernicus.org/articles/10/2275/2018/):

"All *ESSD* papers should have included their specific data link as the final sentence of the abstract and should repeat those links accompanied by all necessary explanation and assistance in the data availability section."

The abstract is modified to include the data link.

- Line 37 Please consider adding a reference to Lazzara et al paper and use more recent State of the Climate reference:
  - Lazzara, M.A., G.A. Weidner, L.M. Keller, J.E. Thom, J.J. Cassano, 2012: Antarctic automatic weather station program: 30 years of polar observations. *Bull. Amer. Meteor. Soc.*, **93**, 1519-1537,.
  - Clem, K. R., S. Barreira, R. L. Fogt, S. Colwell, L. M. Keller, M. A. Lazzara, and D. Mikolajczyk, 2020 (in review): Atmospheric Circulation and surface observations [in "State of the climate in 2019"]. *Bull.Amer. Meteor. Soc.*, 101, S293-W296, doi: 10.1175/BAMS-D-20-0090.1

=> Done

• Line 30-33 - add e.g. to the references- as there are many more...

=> Yes there are more, yet this is a data paper, not a review paper. We already cite 8 references here, we think we should not expand this further.

• Change "manned" to "staffed" in line 42

**=> Done**

• Line 43 - Why is the Ricaud referenced as 2020b when you haven't reference a 2020a?

=> Ricaud et al. (2020a) is cited lines 51, 462 and 501, then referenced line 644

• Line 51 - Capitalization of Layer before ABL

**=> Done**

• Line 76 - Add the word "since" before "grown..."

**=> Done**

• Line 87 has unclear English awkward phrasing

**=> This is tentatively rephrased to make it clearer**

• Minor note to the authors, British Antarctic Survey testing shows less frosting on the nonmarine version of the RM Young than the marine version of the RM Young. This is the sensor used successfully in other AWS networks in Antarctica, with less frosting than the marine version. It is true frosting will continue to happen, but there is little that can be done at remote AWS sites for this problem.

=> Thank you for the note, not sure what to do with it in the paper. In our experience marine and non-marine have the same frosting problems at Dome C, the difference between the 2 models is with the bearings getting stuck at cold temperature in the marine version.

• Line 122 - add "radiation" before "shielding"

**=> Done**

• The averaging of data over 30 minutes does not match WMO standards (see WMO publication #8 for the standards outlined there). This is unfortunate as it can overly smoorh out the wind observations and temperature and pressure less so. However, being on the Antarctic plateau, at the top of Dome Concordia, it is likely the data is not overly impacted, however it should be noted some place (in the metadata especially!) that this is 30 minuteaveraged data and it is not standard practice, as data rarely is averaged over a such a long period.

=> It is now reported that the 30-min average and statistics is not WMO standard practice. The observing system was fully developed by (and for) researchers, and unfortunately did not benefit experienced data / technical support. Following reviewer comment, we will consider adding new data tables to better fit WMO requirements in the future. Please note that 1 minute samples are available but only over 3 years.

• Line 163 - Dome C AWS data is available before 1984 (albeit with gaps)...starting as early as 1980.

**=> OK corrected**

- Lines 170- 175 other AWS are located in this area over the years per poster by Fons et al.. and these other AWS did exist at some point during this time frame....
  - http://amrc.ssec.wisc.edu/outreach/posters/images/fonsAMOMFWposter.pdf

=> Right, the italian AWS is already mentioned and data discussed in the present paper. Some of the authors deployed and operated ,then decommissioned (servicing was too difficult) the 2 stations north and south of Concordia station, which thus no longer operated in the period considered here. Some of the results by these 2 stations, compared with data from the tower, were presented in Genthon et al., 2015. Meteorological and snow accumulation gradients across dome C, east Antarctic plateau, Int. J. Clim., 36, 455-466, DOI: 10.1002/joc.4362.

• Line 203-204 - Is this the warmest ever at every level all on the same day? Would it be over different days for different levels? Perhaps a table of extremes, means, etc with dates would be helpful? This is different than the Appendix table 1 and maybe just a nicety, but would be interesting to document?

=> Not sure how to handle this comment. Lines 203-204 refer to temperature inversion and coldest surface temperature, there is no mention of anything "warmest".

=> Not sure what exactly the suggested additional table should be.

• Figure 5 could use some clarification in title/legend... description is ok...but something that denotes these are differences from the lowest level of the tall tower and these other AWS.

**=> Legend tentatively improved**

• In Figure 7 - am I correct that this is not "model" but observation levels ?

=> Right! Corrected.

• Line 299 - How is any of the data quality controlled? (beyond the winds listed here) This is not addressed.

=> Quality control is addressed lines 148 – 158. This is admittedly fairly limited but we do not know of a more systematic / automatized method to apply. These are research data and we do not think that a "quality disclaimer" is necessary. On the other hand, we expect that any user having doubts on the quality of parts of the data set will raise the issue and we will look into it. Also, see reply on comment by reviewer 2 and corresponding additions in the text.

• So wind is less than ten years....making only temperature a 10 year record (?)

=> Right, so we change the title to: Temperature and wind observations over 10 years on a 45-m tower at Dome C, East Antarctic plateau.

• Line 362 - English phrasing?? Perhaps this sentence needs to be broken up into two different ones to better explain what is happening here. I get the gist of it, but it wasn't clear to me, and I ended up re-reading it several times.

=> Phrase broken into 2 sentences

• Line 396 - say "respectively " at the end of the sentence.

**=> Done**

• Line 430-435: What is the possibility of the ERA5 assimilating Dome C II AWS? (Likely high given it has been used in ERA-I and other reanalysis models...)

=> Not sure we understand. In principle it does, although the analysis procedure may decide to reject some data. Experience suggests that pressure is mostly assimilated but not necessarily wind or temperature.

• Line 470-475 - Minor note - why the funny marks around "coreless", etc.?

=> The funny marks were supposed to be quotes. Corrected. Similar "funny marks" also corrected elsewhere in the text.

• Line 493 - spelling error "tosoverestimate"

**=> Corrected**

• Line 496 - replace "less good agreement" and just say "less agreement"

=> Done

• Line 532 - Was the AMRC data collected quality controlled data used?

=> We understand that these are the quality controlled data. Anyway, the acknowledgments section has been reformulated using phrasing specifically recommended by AMRC and PNRA when using their data.

• Minor note, some references doubled spaced others not ?

=> Right. We use EGU Copernicus template for Microsoft Word to format the paper but actually use Open Office word processor. May be this results from some inconsistency between the 2 word processors, hopefully corrected the at the editing step - if this paper proceeds to editing.

• As a general question - How are the different levels cross calibrated or was that not done for this instrumentation set?

=> There is no cross calibration between levels, only cross checks such as described in the Setting, instruments, data and methods section

• Why talk about other papers not written yet?? Just say what is covered in this paper??

=> We assume that this comment refers to forthcoming papers about atmospheric moisture. The point is to explain and support why we do not report moisture observations in the present paper, although we report that we use thermometer devices that also (tentatively) measure moisture. Moisture measurement need specially adapted sensors which we do not have at 6 levels and over 10 years of observation, thus a specific paper is needed for moisture.

**Anonymous reviewer 2:**

**General**

This paper describes temperature and wind data measured over ten years at six levels on a 42 m tower near Concordia station, which is located on the high interior plateau of the East Antarctic Ice Sheet. The dataset is unique – no other comparable measurements exist from this part of Antarctica – and is thus of immense value for validating the performance of climate and weather prediction models in this region and also for the development and validation of remote sensing techniques. Furthermore, the Dome C region provides an ideal "natural laboratory" for studying the atmospheric boundary layer (ABL) under conditions of strong static stability so these measurements also have application in the development of parametrisations of the ABL under such conditions, which occur widely across the polar regions and even in mid-latitudes. I am very pleased that the authors have chosen to make these data freely available and have documented them carefully in this

paper. The paper provides users of the data with essentially all of the metadata that they would need to access and analyse the data. The description of the quality control procedures applied could possibly be a little more quantitative (see my detailed comments below) but this is not a major issue.

As well as providing a detailed description of the dataset, the paper also presents some basic climatological analysis of the data (section 3) and uses the data to validate the ERA5 reanalysis (section 4). These excellent studies are a substantial part of the paper and would be worthy of publication in their own right in an atmospheric science journal. It is, of course, useful to see examples of the use of the data in a dataset description paper but the Editors may wish to consider whether the balance is right for ESSD. This issue aside, I believe that the paper is suitable for publication following attention to the specific points raised below.

=> We thank the reviewer for insightful and very pertinent comments to improve the paper. Here is how we have taken comments into account and thus hopefully fulfilled the reviewer's requirements. Please note, line numbering is that of the original manuscript, obviously changed in the new manuscript. The new manuscript also accounts for comments by the other reviewer and changes made in response to these comments. Concerning the final point raised above, we think that the comparison is of interest not only to illustrate the value of the observation dataset we make available to the community, but also because meteorological (re)analyses are often used as surrogates for unavailable observation, a view which is particularly questionable for the Antarctic plateau atmospheric boundary layer. We need real observations even when they are hard to obtain.

**Specific points**

Lines 54-56: You could also mention measurements made at Halley (King, 1990, 10.1002/qj.49711649208) and the Alexander Tall Tower! (Mateling et al., 2018, 10.1175/jamc-d-17-0017.1)

=> Although the 2 towers are located on peripheral shelves near sea-level rather than on the high antarctic interior plateau, they have definitely proved very useful to study the peculiarities of the antarctic atmospheric boundary layer. We now mention the towers and list the corresponding references.

2. Lines 92-93: I presume you mean "...upwind of the station in the prevailing wind direction."?

**=> Right, this is reformulated**

3. Line 94: "extension" rather than "expansion"?

**=> OK, taken**

4. Line 138: "mean" rather than "modulus"?

**Actually, a coma was missing which made it confusing, coma now provided**

5. Line 143: "power outages" rather than "blackouts"?

**=> OK, taken**

6. Lines 151-152: These are reasonable limits for quality control purposes but I don't think that it is correct to say that temperature "cannot" be outside these limits.

**=> OK, this is reformulated**

7. Lines 154-155: "Data outside of those ranges, or showing suspicious vertical variations or unrealistically steep changes, are eliminated." Did you apply objective criteria on vertical gradients or temporal changes? If so, state clearly what these are. If this was done subjectively, then make this clear.

=> There is no "objective" data quality analysis and processing in the mathematical sense. On the other hand, he data are 1st subject to automatic filtering (automatic detection / removal or unrealistic profiles, unrealistic steps, etc), completed by visual inspection and manual filtering for particular cases. This is further stated in the paper.

8. Line 155: "rapid" rather than "steep"? (I assume you are referring to temporal change here?)

**=> OK taken**

9. Lines 199-201: This sentence isn't very clear. Are you talking about elevated inversion layers? Not sure what you mean by "nonlinear turbulent diffusion" – turbulence is an inherently nonlinear phenomenon.

=> Yes we are talking about elevated inversion layers, building at the top of the weakly stable boundary layer. Turbulence is indeed nonlinear but we want to emphasize that the increase in inversion strength can only be explained by a non-linear process (Estourneland Guedalia 1985) Linear diffusive processes such as radiation tend to smooth gradients out. We have clarified the sentence as follows:

Strong elevated temperature inversions within the tower height generally build-up at the top of the boundary-layer when non-linear diffusion by turbulence vertically

transports cold air from the surface, e.g. when the stable boundary layer transits from a very stable (with a very strong near surface-based inversion) to a weakly stable regime (Vignon et al. 2017)

10.Figure 3: Use symbols to mark the measurement levels and possibly add horizontal bars to indicate measurement uncertainty. Part (a) shows a suspiciously large superadiabatic temperature gradient between the 25 m and 33 m levels.

=> Done using circle symbol, with diameter reflecting height uncertainty due to snow accumulation during the period of observation

11.Lines 229-230: You could establish how much of the difference between AWS and tower temperature was due to sensor height difference by extrapolating the tower measurements to AWS sensor height (either linearly or using a more sophisticated extrapolation).

=> Yes but the elevation difference varies in time and is not even known at a given time. Considering the combined uncertainties in elevation difference and with extrapolation method, we don't this would bring much additional insight.

12: Figure 6 (and other power spectra). Give units on the "Power" axis.

=> The unit for of power spectrum is the square of the variable  $((^{\circ}C)^2 \text{ or } (m/s)^2)$  multiplied by time sampling (48 samples per day)). This does not make for a very handy unit, so we do not report this on the figures themselves but now mention in the text.

13: Figure 7 caption: "tower level", not "model level".

**=> Done**

14: Figure 9: Y-axis caption needs correction.

**=> Done**

15: Line 328-329, figure 10(a): It is interesting that you don't observe significant power at the inertial frequency.

=> Actually, the spectra do show power near inertial frequency (approx ½ day) but this is very small compared to annual, semi-annual and diurnal. We do not make a case of the inertial waves in the paper because they are intermittent, one would have to "chase" them, and this is beyond the scope of the paper. The text is corrected to mention that gravity waves show on spectra.

16: Section 4: I think it would be useful to summarise the results (mean bias, rms bias, correlation coefficients) of this section in a table, maybe broken down by season?

=> We appreciate this suggestion. We think that graphical representation (figs 12 and 13) is a good way of conveying important information on the agreement between ERA5 and observation. We provide numbers (correlations, rms) on the figure and in the text. Whether this needs to be further summarized in a table is questionable. In her/his general comment, the reviewer wonders if an ESSD paper is the right place for an evaluation of ERA5. We argue that this is used as an illustration of the value of our data but to remain focuses may be we should not expand the presentation of the results further than we currently do (numbers, correlations, rms, are in the text or on the figures, do we need to further summarize in a table in this context?)

17: Lines 373-374: I don't understand this sentence. Correlation coefficients don't tell you anything about the relative amplitudes of variations in the two series that are being correlated.

=> It does tell about the relative amplitude of variations along the series. In terms of timing, the 2 series at https://www.lmd.jussieu.fr/~cgenthon/temporaire/illusrcorr.jpg (sorry for the link, the system here does not allow to directly insert figures) are perfectly coherent in timing of events but the correlation coefficient is less than 1 because the amplitude of the events is not the same.

18: Line 380: The temperature profile has a log-linear form under stable conditions, with the linear component increasing with increasing stability.

=> Right. "linear" changed to "log-linear"

19: Lines 391-392: It might be clearer to say "the reanalysis product has a cold bias at higher temperatures and a large cold bias at the lowest temperatures". Weidner et al. (QJRMS 2021, 10.1002/qj.3901) give a striking example of the failure of ERA-Interim and ERA5 to accurately reproduce an extreme low temperature and associated extreme surface inversion over Greenland.

=> We don't think that this is fully equivalent to what we intend to express, as we also carry a point on the vertical structure of biases. We thus stick to our original formulation.

20: Line 493: Typo "to overestimate".

**=> Corrected**

21: References, line 573: I don't think that the Ekman paper is referenced anywhere in the text.

**=> Right. Reference removed**

**A note on the datasets**

The datasets described in the paper can be easily downloaded from the Pangaea data centre as tabdelimited text files with some basic metadata provided in the file headers. Temperature and wind speed measurements are provided in separate files. However, wind direction measurements (which are described in the paper) are not made available. As wind direction is a cyclic variable there can be some ambiguity (alluded to in the paper) when calculating 30-minute means, but I would strongly encourage the authors to deposit a third file in Pangaea, containing either 30-minute vector mean wind directions or, alternatively, 30-minute mean u- and v-components of the wind.

=> This is a good suggestion. However PANGAEA currently informs that it is overwhelmed with data accommodation requests and is very slow to answer. Concerning the the temperature and wind speed data for this paper, waiting for PANGAEA's DOIs has significantly delayed the process of paper submission. Waiting for PANGAEA to approve and provide DOI for an additional data set would further delay acceptance and publication by several months. We will consider depositing 30-minute U- and V- components in the future but in order to not delay further the paper we do not advertise this in the paper. On the other hand, at the same time we mention that the minute instant directions shown in the paper are available on request to authors, we add that the 30-minute average U and V are also available on request.